# Understanding the Effect of Conserved Charges on the Coalescence Sum Rule of Directed Flow

Kishora Nayak [1],* , Shu-Su Shi [2] and Zi-Wei Lin [3]

1 Department of Physics, Panchayat College, Sambalpur University, Bargarh 768028, India
2 Key Laboratory of Quark & Lepton Physics (MOE) and Institute of Particle Physics, Central China Normal University, Wuhan 430079, China
3 Department of Physics, East Carolina University, Greenville, NC 27858, USA; linz@ecu.edu
* Correspondence: k.nayak1234@gmail.com

**Abstract:** Recently, the rapidity-odd directed flow ($v_1$) of produced hadrons ($K^-$, $\phi$, $\overline{p}$, $\overline{\Lambda}$, $\overline{\Xi}^+$, $\Omega^-$, and $\overline{\Omega}^+$) has been studied. Several combinations of these produced hadrons, with very small mass differences but differences in the net electric charge ($\Delta q$) and net strangeness ($\Delta S$) on the two sides, have been considered. A difference in $v_1$ between the two sides of these combinations ($\Delta v_1$) has been proposed as a consequence of the electromagnetic field produced in relativistic heavy-ion collisions, especially if $\Delta v_1$ increases with $\Delta q$. Our study is performed to understand the effect of the coalescence sum rule (CSR) on $\Delta v_1$. We point out that the CSR leads to $\Delta v_1 = c_q \Delta q + c_S \Delta S$, where the coefficients $c_q$ and $c_S$ reflect the $\Delta v_1$ of produced quarks. Equivalently, one can write $\Delta v_1 = c_q \Delta q + c_B \Delta B$, involving the difference in the net baryon number $\Delta B$, where the CSR gives $c_B = -3c_S$. We then propose two methods to extract the coefficients for the $\Delta q$ and $\Delta S$ dependences of $\Delta v_1$.

**Keywords:** directed flow; coalescence sum rule





## 1. Introduction

The properties of strongly interacting quark-gluon plasma, produced by relativistic heavy-ion collisions, can be studied using anisotropic flows, including the directed flow ($v_1$) [1–3]. $v_1$ has been found to be a sensitive probe of the equation of state of the produced medium [4,5]. The anisotropic flows of identified hadrons are expected to follow the coalescence sum rule (CSR) when the produced matter is initially in parton degrees of freedom and hadronizes via quark coalescence. The CSR states that the sum of constituent quarks' flow equals the corresponding flow of the hadron; it leads to the number-of-constituent-quark scaling and provides information about the particle production mechanism and partonic collectivity [6,7]. This paper is based on our recent study [8], which was motivated by Ref. [9].

A new method of testing the coalescence sum rule using the difference in $v_1$ in different combinations of produced hadrons [$K^-$ ($\bar{u}s$), $\phi(s\bar{s})$, $\overline{p}(\bar{u}\bar{u}\bar{d})$, $\overline{\Lambda}(\bar{u}\bar{d}\bar{s})$, $\overline{\Xi}^+$ ($\bar{d}\bar{s}\bar{s}$), $\Omega^-$ ($sss$) and $\overline{\Omega}^+$ ($\bar{s}\bar{s}\bar{s}$)] has been previously proposed [9]. These selected hadrons are all produced, consisting of $\bar{u}$, $\bar{d}$, $s$, and $\bar{s}$ quarks. In contrast to the produced hadrons, transported hadrons such as $\pi^\pm$, $p$, and $\Lambda$ receive contributions from initial-state (incoming) $u$ and $d$ quarks, along with the produced quarks, which complicates the interpretations of the CSR. Table 1 shows a selection of five independent hadron sets [8]. Sets 1–3 are identical, with the same quark contents on both sides, whereas sets 4–5 are non-identical. Indeed, different choices of five independent hadron sets can be made [8]. One can also obtain all ten sets from Table 1 in Ref. [9]; however, they are not all independent. In a given set, the difference in $v_1$ between the left and right sides (after including the weighting factors of each hadron as listed in Table 1) is termed $\Delta v_1$. Similarly, the differences in the net electric charge ($\Delta q$ or $\Delta q_{ud}$) and net strangeness ($\Delta S$) are obtained.

As $v_1$ develops in the early stage of collisions, it is sensitive to the strong electromagnetic field produced by incoming protons in the two colliding nuclei. The correlation between the difference in $v_1$ on $\Delta q$ has been considered as breaking the coalescence sum rule and being a consequence of the electromagnetic fields [9,10], especially when the difference in $v_1$ increases with $\Delta q$. This motivated us to critically examine the consequence of the coalescence sum rule on the difference in $v_1$ [8]. This paper is organized as follows. The derivation of the CSR relation is given in Section 2. Two methods for extracting the dependence of $\Delta v_1$ on $\Delta q$ and $\Delta S$ are presented in Section 3. A summary is finally given in Section 4.

## 2. The CSR Relation for the Difference in $v_1$ of a Hadron Set

The relation between the $v_1$ of a hadron (H) and those of its constituent quarks from the CSR can be written as

$$v_{1,\mathrm{H}}(p_{\mathrm{T,H}}) = \sum_j v_{1,j}(p_{\mathrm{T},j}), \qquad (1)$$

which is a sum over each constituent quark $j$ of the hadron. The simplest case for the coalescence of comoving equal-mass quarks gives the usual relation [7] $v_{1,\mathrm{H}}(N_{cq}\, p_{\mathrm{T,q}}) = N_{cq}\, v_{1,q}(p_{\mathrm{T},q})$, where $N_{cq}$ is the number of constituent quarks of the hadron.

For each of the five hadron sets in Table 1, the number of quarks of flavor $i$ ($N_i$) for each side (L: left; R: right) is calculated as the sum of the number of quarks of flavor $i$ in each hadron, multiplied by the weighting factor. With $\Delta N_i = N_i^L - N_i^R$, each set satisfies the following relations:

$$\Delta N_{\bar{u}} + \Delta N_{\bar{d}} = 0, \ \ \Delta N_s + \Delta N_{\bar{s}} = 0. \qquad (2)$$

In other words, the two sides have the same number of $\bar{u} + \bar{d}$ quarks and the same number of $s + \bar{s}$ quarks. However, they may have different $N_{\bar{u}}$ and/or $N_s$. Therefore, they can have a different total net electric charge ($q$), total net strangeness ($S$), or total net electric charge in light quarks ($q_{ud}$), defined as $\Delta q = q^L - q^R$, $\Delta S = S^L - S^R$, and $\Delta q_{ud} = q_{ud}^L - q_{ud}^R$, respectively. For the hadron sets in Table 1, Equation (2) leads to

$$\Delta q_{ud} = \Delta N_{\bar{d}}, \ \ \Delta S = 2\Delta N_{\bar{s}}, \ \ \Delta q = \Delta q_{ud} + \Delta S/3. \qquad (3)$$

**Table 1.** List of five independent hadron sets, including identical ($\Delta q = 0$ and $\Delta S = 0$) and non-identical ($\Delta q \neq 0$ and/or $\Delta S \neq 0$) constituent quark combinations with similar mass.

| Set | $\Delta q$ | $\Delta q_{ud}$ | $\Delta S$ | $\Delta B$ | Left Side (L) | Right Side (R) |
|-----|-----|-----|-----|-----|-----|-----|
| 1 | 0 | 0 | 0 | 0 | $v_1[K^-(\bar{u}s)] + v_1[\overline{\Lambda}(\bar{u}\bar{d}\bar{s})]$ | $v_1[\bar{p}(\bar{u}\bar{u}\bar{d})] + v_1[\phi(s\bar{s})]$ |
| 2 | 0 | 0 | 0 | 0 | $v_1[\overline{\Lambda}(\bar{u}\bar{d}\bar{s})]$ | $\frac{v_1}{2}[\overline{\Xi}^+(\bar{d}\bar{s}\bar{s})] + \frac{v_1}{2}[\bar{p}(\bar{u}\bar{u}\bar{d})]$ |
| 3 | 0 | 0 | 0 | 0 | $\frac{v_1}{3}[\Omega^-(sss)] + \frac{v_1}{3}[\overline{\Omega}^+(\bar{s}\bar{s}\bar{s})]$ | $v_1[\phi(s\bar{s})]$ |
| 4 | 1/3 | 0 | 1 | −1/3 | $\frac{v_1}{2}[\phi(s\bar{s})]$ | $\frac{v_1}{3}[\Omega^-(sss)]$ |
| 5 | 2/3 | 1/3 | 1 | −1/3 | $\frac{v_1}{2}[\phi(s\bar{s})] + \frac{v_1}{3}[\bar{p}(\bar{u}\bar{u}\bar{d})]$ | $v_1[K^-(\bar{u}s)]$ |

The total $v_1$ from each side of a hadron set can be written as $v_1^{L,R} = \sum_i N_i^{L,R}\, v_{1,i}$. It is important to note that we neglect the effect of different constituent quark masses in this study. Therefore, the above relation applies to light and strange (anti)quarks at the same $p_{\mathrm{T}}$ value or $p_{\mathrm{T}}$ range, and consequently, it also applies to baryons at a $p_{\mathrm{T}}$ value or range that is $3/2$ times that of the meson's. The difference in $v_1$ between the two sides is then given by

$$\begin{aligned}
\Delta v_1 &= v_1^L - v_1^R = \sum_i \Delta N_i\, v_{1,i} \\
&= (v_{1,\bar{d}} - v_{1,\bar{u}})\Delta q_{ud} + \left(\frac{v_{1,\bar{s}} - v_{1,s}}{2}\right)\Delta S \qquad (4) \\
&= \left(v_{1,\bar{d}} - v_{1,\bar{u}}\right)\Delta q + \left(\frac{v_{1,\bar{s}} - v_{1,s}}{2} - \frac{v_{1,\bar{d}} - v_{1,\bar{u}}}{3}\right)\Delta S. \qquad (5)
\end{aligned}$$

Equations (4) and (5) show the linear dependence of $\Delta v_1$ of a hadron set on both $\Delta q$ and $\Delta S$, where the coefficients are simply determined by the quark-level differences in $v_1$. We can also see that the coefficient for the $\Delta S$ dependence in Equation (5) is not as clean as the corresponding coefficient in Equation (4).

The difference in $v_1$ in the hadron sets also depends on $\Delta B$, which represents the difference in the net baryon number between the two sides of a hadron set. Under the condition from Equation (2), we have

$$\Delta B = -2\Delta N_{\bar{s}}/3 = -\Delta S/3. \tag{6}$$

Therefore, $\Delta v_1$ depends linearly on both $\Delta q$ and $\Delta B$ as follows:

$$\Delta v_1 \;=\; (v_{1,\bar{d}} - v_{1,\bar{u}})\Delta q_{ud} - 3\left(\frac{v_{1,\bar{s}} - v_{1,s}}{2}\right)\Delta B \tag{7}$$

$$\;=\; \left(v_{1,\bar{d}} - v_{1,\bar{u}}\right)\Delta q - 3\left(\frac{v_{1,\bar{s}} - v_{1,s}}{2} - \frac{v_{1,\bar{d}} - v_{1,\bar{u}}}{3}\right)\Delta B. \tag{8}$$

Assuming that the rapidity of a hadron formed by quark coalescence is the same as that of the coalescing quarks, the $v_1$ slope at $y = 0$, denoted as $v_1' = dv_1/dy(y = 0)$, satisfies exactly the same relations as Equations (4), (5), (7) and (8), where one just needs to replace $v_1$ with $v_1'$ [8].

### 3. Extracting the $\Delta q$ and $\Delta S$ Dependences of the Coefficients

The coalescence sum rule may not be satisfied for certain collision energies and/or systems, e.g., in cases where $v_1$ is not dominated by parton dynamics. Since Equations (4) and (5) from the coalescence sum rule predict $\Delta v_1 = 0$ for hadron sets with $\Delta S = \Delta q_{ud} = 0$ or $\Delta S = \Delta q = 0$, we propose that the following modified equations are used to fit the $\Delta v_1$ data from the five independent sets:

$$\Delta v_1 \;=\; c_0 + c_q\,\Delta q_{ud} + c_S\,\Delta S, \tag{9}$$

$$\Delta v_1 \;=\; c_0^* + c_q^*\,\Delta q + c_S^*\,\Delta S. \tag{10}$$

This way, a non-zero value of the intercept parameter, denoted as $c_0$ or $c_0^*$, would mean breaking the coalescence sum rule. Note that even for a given collision system, the coalescence sum rule may be satisfied around midrapidity but not satisfied at large rapidities; therefore, these coefficients are rapidity-dependent.

To extract the coefficients of the $\Delta q$ and $\Delta S$ dependences of $\Delta v_1$ for a given collision system, one can utilize the five-set method by simply fitting the 5 data points using Equation (9) or Equation (10). It should be noted that the fit function represents a two-dimensional plane over the $\Delta q - \Delta S$ space. Therefore, one should not simply fit the $\Delta v_1$ data with a one-dimensional function of $\Delta q$ without taking into account the different $\Delta S$ values of different hadron sets.

Alternatively, we can use the three-set method [8], where we take the average of sets 1–3, because they have the same $\Delta q$ and $\Delta S$ values, and denote the average as set A. Let us also denote set 4 and set 5 as set B and set C, respectively. Equation (9) then leads to $(\Delta v_1)_A = c_0$, $(\Delta v_1)_B = c_0 + c_S$, and $(\Delta v_1)_C = c_0 + c_q/3 + c_S$. Thus, we obtain

$$c_0 = (\Delta v_1)_A; \;\; c_q = 3[(\Delta v_1)_C - (\Delta v_1)_B]; \;\; c_S = (\Delta v_1)_B - (\Delta v_1)_A. \tag{11}$$

Similarly, the three-set method yields the coefficients in Equation (10) as

$$c_0^\star = (\Delta v_1)_A = c_0; \;\; c_q^\star = 3[(\Delta v_1)_C - (\Delta v_1)_B] = c_q;$$
$$c_S^\star = 2(\Delta v_1)_B - (\Delta v_1)_A - (\Delta v_1)_C = c_S - c_q/3. \tag{12}$$

We see that the three-set method has an advantage over the five-set method in that we can extract the $\Delta q$ and $\Delta S$ coefficients analytically instead of performing a two-dimensional fit. It should be noted that the CSR predicts the following coefficients [8]:

$$c_0 = c_0^* = 0; \quad c_q = c_q^\star = v_{1,\bar{d}} - v_{1,\bar{u}}; \tag{13}$$

$$c_S = (v_{1,\bar{s}} - v_{1,s})/2, \quad c_S^\star = (v_{1,\bar{s}} - v_{1,s})/2 - (v_{1,\bar{d}} - v_{1,\bar{u}})/3. \tag{14}$$

When the difference in $v_1$ of the hadron sets is expressed in terms of $\Delta B$, we can use the following modified equations, which are similar to Equations (9) and (10):

$$\Delta v_1 = c_0 + c_q \, \Delta q_{ud} + c_B \, \Delta B, \tag{15}$$

$$\Delta v_1 = c_0^* + c_q^* \, \Delta q + c_B^* \, \Delta B. \tag{16}$$

Here, the CSR predicts the following $\Delta B$ coefficients:

$$c_B = -3(v_{1,\bar{s}} - v_{1,s})/2 = -3\,c_S, \quad c_B^\star = -3(v_{1,\bar{s}} - v_{1,s})/2 + (v_{1,\bar{d}} - v_{1,\bar{u}}) = -3\,c_S^\star. \tag{17}$$

Obviously, the CSR relates the $\Delta q$ and $\Delta S$ (or $\Delta B$) coefficients to the quark-level differences in $v_1$. It should be noted that the quark $v_1$ in Equations (13), (14) and (17) refers to its value after the partonic evolution just before the quark coalescence. Therefore, these coefficients could be non-zero due to the flavor dependence of the strong interaction or the effect of the electromagnetic field on the produced quarks. Additionally, the extraction methods outlined in this section apply to the $v_1'$ data in exactly the same way. Furthermore, in the full study [8], we demonstrated these extraction methods for the $\Delta q$ and $\Delta S$ coefficients using results from the AMPT model for Au+Au collisions at different energies.

## 4. Summary

In this study, we derived the relations for $\Delta v_1$, representing the difference in $v_1$ between the two sides of a hadron set, using the coalescence sum rule. In earlier studies, seven produced hadron species ($K^-$, $\phi$, $\overline{p}$, $\overline{\Lambda}$, $\overline{\Xi}^+$, $\Omega^-$, and $\overline{\Omega}^+$) were considered, and a non-zero $\Delta v_1$ dependence on the difference in the net electric charge ($\Delta q$) was considered as breaking the coalescence sum rule and being a consequence of the electromagnetic fields. Our study showed that the coalescence sum rule only leads to a zero $\Delta v_1$ for a hadron set if its two sides have identical constituent quark contents (i.e., $\Delta q = \Delta S = 0$). In general, $\Delta v_1$ depends linearly on both $\Delta q$ and $\Delta S$, or on both $\Delta q_{ud}$ and $\Delta S$, and the same applies to the difference in the $v_1$ slopes at midrapidity ($\Delta v_1'$). Since there are only five such independent hadron sets, there will be five independent data points from the measurement of a given collision system. We propose using a two-dimensional function such as $c_0 + c_q \Delta q_{ud} + c_S \Delta S$ to extract the coefficients of the $\Delta q_{ud}$ and $\Delta S$ dependences, where a non-zero value of the coefficient $c_0$ indicates the breaking of the coalescence sum rule. In the five-set method, one fits the five data points using this function. On the other hand, in the three-set method, the three data points with the same $\Delta q$ and $\Delta S$ values are combined into one set, allowing us to obtain the coefficients analytically. The coalescence sum rule relates these coefficients to the quark-level differences in $v_1$ just before quark coalescence. Therefore, the coefficients can be affected by both the strong interaction and the electromagnetic fields. Equivalently, one can also express $\Delta v_1$ or $\Delta v_1'$ in terms of the net baryon difference $\Delta B$, where the function $c_0 + c_q \Delta q_{ud} + c_B \Delta B$ can be used to extract the coefficients, and the coalescence sum rule yields $c_B = -3c_S$.

**Author Contributions:** Conceptualization, Z.-W.L.; Methodology, Z.-W.L.; Software, K.N.; Formal analysis, K.N.; Resources, S.-S.S.; Writing—original draft, Z.-W.L.; Writing—review and editing, K.N.; Visualization, K.N.; Supervision, Z.-W.L.; Project administration, S.-S.S.; Funding acquisition, S.-S.S. All authors have read and agreed to the published version of the manuscript.

**Funding:** S.S. was supported in part by the National Key Research and Development Program of China under Grant Nos. 2022YFA1604900 and 2020YFE0202002, and the National Natural Science

Foundation of China under Grant Nos. 12175084 and 11890710 (11890711). K.N. was supported by OSHEC, Department of Higher Education, Government of Odisha, Index No. 23EM/PH/124, under MRIP 2023. Z.-W.L. was supported by the National Science Foundation under Grant Nos. 2012947 and 2310021. The APC was funded by the National Science Foundation under Grant No. 2012947.

**Data Availability Statement:** Data are contained within the article.

**Acknowledgments:** We thank Sandeep Chatterjee for the discussions.

**Conflicts of Interest:** The authors declare no conflict of interest.

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
