# Peer review of "Understanding the Effect of Conserved Charges on the Coalescence Sum Rule of Directed Flow"

_universe, doi:10.3390/universe10030112_

Round 1
Reviewer 1 Report
Comments and Suggestions for Authors
The presented is work is dedicated to studies of mechanisms of directed flow in A+A collisions using the AMPT model.
The main observation provided by the authors is that for the lowest considered collision energy after the fitting procedure they get non-zero value of the intercept parameter c0. Authors claim that this observation means that the coalescence sum rule is broken in this case.
Isn't it because for the lowest collision energy EM effects reach also midrapidty to some extent?
For larger collision energies delta V is more or less flat in rapidity in the presented interval -1.5<y<1.5 and all extracted parameters are close to 0. But in the Monte Carlo simulations one can extend the rapidity range and instead of fitting dV/dy at y=0 find the point where deltaV deviates from 0. I expect this point to be collision energy dependent. In this way one should be able to find the kinematic region where the coalescence sum rule breaks.
In other words my main concern is with the equations 6 and 7. It seems that parameters c should be made rapidity-dependent there.
Minor questions:
1) Why was AMPT selected for these studies?
2) Why was 10-50% centrality class used?
3) How large was the MC statistics?
4) Figure 3 - what is the quality of the fits?
I believe it would be useful to add answers to these questions to the paper.
Minor editing comments:
abstract - delta V quantity is not defined in the abstract. I would suggest to extend it with a couple of words indicating what this quantity is. Otherwise reader would have to wait for a whole page while it appears
table 1 - brackets should be corrected for phi, v is missing for Omega etc.
figure 1 - label '200 gev' is missing for the central bottom plot
Reviewer 2 Report
Comments and Suggestions for Authors
Reply to Author(s)
Title: Understanding the Effect of Strangeness and Electric Charge on 4 Coalescence Sum Rule of Directed Flow
The authors have tried to address the effect of strangeness(S) and electric charge(q) on the Coalescence Sum Rule (SCR) of directed flow (v1) using the AMPT MC model. The study is performed to understand these effects in the absence of electromagnetic fields. There are some issues which need to be addressed (very critical) before going for the publications.
Major comments:
-
The authors in this article have cited Ref. 8, an article written by the same authors and share similar results. The author should clarify whether the previous study is submitted to any other journals or not, as it is not clear while reading the text.
-
As we can see from previous studies at lower energies the AMPT model does not describe qualitatively the v1 measurements. The authors should explicitly comment on this.
-
The author found a nonzero value of the first coefficient (c0) at lower energies, which indicates the breaking of the coalescence sum rule, however the author should discuss the physics origin rather than just the observation.
-
Also the author should address in the text on the physics origin on the dependency of extracted parameters on different choices of the hadron sets (Fig. 4) used in this study.
Minor comments:
Line 19: abscence → absence
Equation 2: Please correct the indices
Fig 2: Please add the energy legend for 200 AGeV
Round 2
Reviewer 1 Report
Comments and Suggestions for Authors
I have no comment to the shortened version of the paper. Now it looks consistent and all the conclusions are supported by the main part.